# Comparative Expression Profiling Reveals the Regulatory Effects of Dietary Mannan Oligosaccharides on the Intestinal Immune Response of Juvenile *Megalobrama amblycephala* against *Aeromonas hydrophila* Infection

**DOI:** 10.3390/ijms24032207

**Published:** 2023-01-22

**Authors:** Xiaoheng Zhao, Xu Wang, Hong Li, Yunlong Liu, Yancui Zheng, Hongping Li, Minying Zhang, Hanliang Cheng, Jianhe Xu, Xiangning Chen, Zhujin Ding

**Affiliations:** 1Jiangsu Key Laboratory of Marine Bioresources and Environment, Co-Innovation Center of Jiangsu Marine Bio-Industry Technology, Jiangsu Ocean University, Lianyungang 222005, China; 2Jiangsu Key Laboratory of Marine Biotechnology, School of Marine Science and Fisheries, Jiangsu Ocean University, Lianyungang 222005, China; 3Hunan Fisheries Science Institute, Changsha 410153, China

**Keywords:** mannan oligosaccharides, *Megalobrama amblycephala*, DGE profiling, differentially expressed genes, intestinal immunity, intestinal barrier

## Abstract

Mannan oligosaccharides (MOS) are functional oligosaccharides with beneficial effects on the non-specific immunity of *Megalobrama amblycephala*, but systematic studies on the immunomodulatory mechanisms of MOS are still lacking. To investigate the protective mechanisms of three different levels of dietary MOS supplementation on the intestinal immunity of juvenile *M. amblycephala*, comparative digital gene expression (DGE) profiling was performed. In this study, 622 differentially expressed genes (DEGs) were identified, while the similar expression tendency of 34 genes by qRT-PCR validated the accuracy of the DGE analyses. Gene Ontology (GO) enrichment revealed that the DEGs were mainly enriched in two functional categories of biological process and molecular function. Kyoto Encyclopedia of Genes and Genomes (KEGG) analysis revealed that the DEGs were mainly related to complement and coagulation cascades, coagulation cascades, platelet activation, natural killer cell mediated cytotoxicity, Fc gamma R-mediated phagocytosis and antigen processing and presentation. In addition, the pro-inflammatory, apoptosis and tight junction-related genes were more significantly up-regulated upon infection in the dietary MOS groups to enhance host immune functions and maintain the stability of the intestinal barrier. These results will be helpful to clarify the regulatory mechanism of MOS on the intestinal immunity of *M. amblycephala* and lay the theoretical foundation for the prevention and protection of fish bacterial diseases.

## 1. Introduction

In the past decades, antibiotics have been supplemented to aquatic feeds to enhance host immunity and thus improve the survival rates of aquatic animals. It has also been found that peptide antibiotics can improve the weight gain rate and feed efficiency of aquatic animals. However, as the problem of antibiotic residues becomes more serious, the development of the aquaculture industry is restricted, thus the application of antibiotics in aquatic feed has been banned or reduced.

According to their biological functions, oligosaccharides can be divided into common oligosaccharides and functional oligosaccharides [1]. The common oligosaccharides can be digested and absorbed by the host and provide energy, while the functional oligosaccharides are not easily digested and absorbed, but have specific biological effects. Compared to antibiotics, functional oligosaccharides are free from bacterial resistance and drug residues, which are able to reach the intestine without inactivation [2]. Therefore, functional oligosaccharides are considered to be an ideal feed additive to promote animal growth and resistance to infection. There are more than a thousand known functional oligosaccharides, while only a few of them are of natural origin. Chitosan oligosaccharides, mannan oligosaccharides, fructo-oligosaccharides, soya oligosaccharides, xylose oligosaccharides, isomalto-oligosaccharides and galacto-oligosaccharides are the main functional oligosaccharides that are currently used as feed additives [3].

Mannan oligosaccharides (MOS), also known as mannooligosaccharides, are a complex of phosphorylated glucose and mannose, and mainly produced by enzymatic digestion which is widely found in plant polysaccharides and the cell walls of many organisms [4]. The MOS currently used in animal feeds are mainly extracted from yeast cell walls, with the main chain commonly linked by α-1,6 glycosidic bonds and the side chains mainly linked by α-1,2 and α-1,3 glycosidic bonds [5]. MOS from different sources possess various structures and physicochemical properties. MOS are usually soluble in water and insoluble in organic solvents, such as ethanol and acetone, and are stable under general feed processing conditions.

MOS have been widely used as feed additives in livestock and poultry farming, which has also been found to be effective on aquatic animals. Studies have shown that dietary MOS increase the weight gain rates of *Oreochromis niloticus* [6], *Sparus aurata* [7], *Dicentrarchus labrax* [8], *Lates calcarifer* [9] and *Pangasianodon hypophthalmus* [10], which may be affected by the appropriate supplemental amount of dietary MOS. This is consistent with our previous study, which found that the supplementation of 200 mg/kg of dietary MOS increased the feed utilization and weight gain rate of juvenile *Megalobrama amblycephala*, but not the 400 mg/kg group [11].

MOS also improve fish immunity, which can enhance host resistance to pathogenic bacterial infections [12]. Dietary MOS as an immune enhancer or feed additive mainly interact with the intestines, thus providing immune protection to the host, and studies are mainly focused on their effects on intestinal histomorphology, microbial system and immunity [13]. Dietary MOS increase the length of intestinal villi and thickness of the muscle layer in Japanese eel (*Anguilla japonica*) [14]. Similarly, the supplementation of dietary MOS increases the intestinal villi length of juvenile white seabream (*Diplodus sargus*), thus the absorption surface area of the intestine [4]. Our previous study also found that the supplementation of 200 mg/kg dietary MOS increased the height and density of intestinal villi of juvenile *M. amblycephala* [11].

Except for affecting the morphology of the fish intestines, MOS also regulate the host intestinal micro-ecological system. The pathogenic bacteria can bind to the intestinal epithelium via specific mannose lectins or other transmembrane proteins, while the presence of MOS in host intestines can reduce bacterial colonization in the intestines and thus adjust the intestinal micro-ecological system [5,15]. The supplementation of appropriate amounts of dietary MOS can promote the proliferation of probiotics in the intestines of juvenile *M. amblycephala*, ensuring the intestinal micro-ecological balance and the homeostasis of the internal environment of digestive tracts, which in turn exhibited a better protective effect against pathogenic bacteria invasion [11]. In addition, dietary MOS can increase the number of intestinal goblet cells and reduce the colonization of pathogenic bacteria, thereby synthesizing and secreting mucins that form a mucosal barrier to protect epithelial cells [16]. However, little is known about the regulatory mechanisms of MOS to the intestinal immunity of aquatic animals.

*M. amblycephala* belongs to Cypriniformes, Cyprinidae, which is one of the major freshwater aquaculture species in China. However, in recent years, a gradual decline of disease resistance and the frequent occurrence of bacterial diseases have been found in *M. amblycephala* aquaculture, among which bacterial septicemia caused by the *Aeromonas hydrophila* infection is the most serious and has become the main factor limiting the sustainable development of the *M. amblycephala* aquaculture [17]. Fish intestines have been proven to be one of the main infection routes of *A. hydrophila*, where it proliferates and penetrates the intestinal mucosal barrier, subsequently infecting other parts of the fish and causing septicemia. As lower vertebrates, the specific immune system of fish is not yet well developed and the innate immunity plays an important role in defense against disease. Thus, the systematic study of intestinal immunity is of great importance for the control of fish bacterial diseases.

Digital gene expression (DGE) profiling can provide a holistic understanding of the gene expression and signaling pathways upon infection [18,19], thus laying the foundation for further explanation of the immune defense mechanisms of fish immunity. In this study, DGE profiling was performed to analyze the regulatory effects of MOS on the intestinal immune signaling of juvenile *M. amblycephala* upon *A. hydrophila* infection. The results identified differentially expressed genes and signaling pathways that are involved in inflammation, apoptosis and tight junction in *M. amblycephala* intestines after dietary MOS supplementation and *A. hydrophila* infection. Thereby, the underlying regulatory signaling was elucidated, which can provide a theoretical basis for fish disease prevention.

## 2. Results

### 2.1. Sequencing Quality, Reads Mapping and Annotation

In order to investigate the immune protective mechanism of MOS on the intestines of juvenile *M. amblycephala* and the dynamic gene expression patterns of immune-related genes in response to *A. hydrophila* infection, the DGE profiling of 27 intestinal samples from the control, MOS200 and MOS400 groups at 0, 4 and 24 hpi were examined using the DNBSEQ sequencing platform and the low-quality reads were filtered. As shown in Table 1, the raw reads obtained in the intestinal samples of *M. amblycephala* ranged from 20.35 M to 21.75 M, which are available in the NCBI database (Accession No. PRJNA846987). The Q20 and Q30 values were greater than 97.75% and 93.27%, respectively, with the error rates less than 0.04% and GC contents approximately 46.19%. The obtained clean reads after data filtering ranged between 20.07–21.5 M, and the average mapping rate of clean reads mapped to the *M. amblycephala* genome was 92.04%, with a total of 22, 875 genes identified.

### 2.2. Identification and Statistics of Differentially Expressed Genes (DEGs)

The DEGs were identified by comparing the MOS treated groups with the control group at different time points upon infection, and the statistics are shown in Figure 1 and Table 2. A total of 622 DEGs (FDR ≤ 0.05 and |log2(fold-change)| ≥ 1), including 198 up-regulated and 434 down-regulated genes, were obtained after the removal of duplicates. Compared with the control group at 0 hpi, the MOS200 and MOS400 groups identified 168 DEGs (93 up-regulated and 75 down-regulated) and 155 DEGs (46 up-regulated and 109 down-regulated), respectively (*p* < 0.05). The number of identified DEGs post-*A. hydrophila* infection (4 and 24 hpi) was less than that at 0 hpi, and it was much less at 4 hpi in both MOS treated groups, especially that of the MOS200 group with only 13 DEGs. Additionally, it was worth noting that the down-regulated DEGs were much more than the up-regulated DEGs in most cases, except for that of the MOS200 vs. control group at 0 hpi.

### 2.3. Functional Annotation of DEGs

To further investigate the dynamic functions of the identified DEGs and the underlying signaling of juvenile *M. amblycephala* against *A. hydrophila* infection, Gene Ontology (GO) and Kyoto Encyclopedia of Genes and Genomes (KEGG) enrichment were performed for functional annotation. As shown in Table 2, the enriched number of DEGs in the GO and KEGG databases was 176 and 570, respectively, which showed that most of the identified DEGs could be annotated in KEGG.

According to the functional annotation by GO annotation, 622 identified DEGs were categorized into three functional categories: biological process, cellular component and molecular function, which showed similar patterns of enrichment and were mainly enriched in the biological process and molecular function at different time points (Figure 2). Specifically, in the category of biological process, DEGs were mainly centered on cellular process, biological regulation, metabolic process and response to stimulus and were like the functional groups, which included enriched GO terms such as the regulation of cellular/biological/cellular biosynthetic processes, regulation of gene expression, regulation of primary/cellular/macromolecule/nitrogen compound metabolic processes and the response to stress/chemical. DEGs enriched in the cellular component were mainly concentrated in the cellular anatomical entity and intracellular-associated GO terms, including the membrane, membrane-bounded organelle, integral/intrinsic component of membrane, extracellular region and space, and cytoplasm, which might relate to the structures or functions of intestinal epithelial cells. In the molecular function term, DEGs were mainly enriched in the binding and catalytic activity associated with the GO terms, such as protein binding, carbohydrate derivative binding, anion binding, small molecule binding and oxidoreductase activity, indicating that MOS might play an important role in protecting the intestinal barrier by maintaining the barrier structures and functions.

The 622 DEGs were further functionally annotated according to the KEGG database and categorized in terms of cellular processes, environmental information processing, genetic information processing, metabolism and organismal systems, which were mainly enriched in the metabolism and organismal systems (Figure 3). The analysis revealed that the DEGs enriched in organismal systems were mainly in the digestive system and immune system, with the enriched digestive system mainly including fat digestion and absorption (map04975), cholesterol metabolism (map04979), bile secretion (map04976), protein digestion and absorption (map04974) and pancreatic secretion (map04972). Moreover, the enriched immune system mainly included complement and coagulation cascades (map04973), coagulation cascades (map04610), platelet activation (map04611), natural killer cell mediated cytotoxicity (map04650), Fc gamma R-mediated phagocytosis (map04666) and antigen processing and presentation (map04612).

The top 10 enriched KEGG pathways (FDR ≤ 0.01) of the identified DEGs at different time points post-infection were shown in Table 3. At 0 hpi, the DEGs identified by a comparison of the MOS200/MOS400 groups with the control group were mainly enriched in the KEGG pathways of the complement and coagulation cascades (map04610), PPAR signaling pathway (map03320), cholesterol metabolism (map04979) and several other digestive-related pathways, indicating that dietary MOS supplementation for 8 weeks could effectively regulate host digestive and immune systems, thereby affecting the feed utilization and immunity of juvenile *M. amblycephala*.

At 4 hpi, the identified DEGs of the MOS200 group vs. the control group were few, thus they were not dramatically enriched. Notably, both the DEGs of the MOS400 vs. the control group at 4 hpi and the MOS200 vs. the control group at 24 hpi were mainly enriched in the pathways of pancreatic secretion (map04972), natural killer cell-mediated cytotoxicity (map04650), protein digestion and absorption (map04974) and apoptosis (map04210), indicating that dietary MOS also affected the nutrient digestion, physiological function and survival of *M. amblycephala*, and the regulatory effect of the MOS200 group was later than that of the MOS400 group. The DEGs of the MOS400 vs. the control group at 24 hpi were mainly enriched in the pathways of lysosome (map04142), phagosome (map04145) and antigen processing and presentation (map04612), which were mainly host immune response related, revealing that dietary MOS (400 mg/kg) significantly enhanced the anti-infection ability of *M. amblycephala*.

### 2.4. Regulation of MOS on Complement-Related Genes

A total of 92 complement-related DEGs were identified between dietary MOS groups and the control group at various time points post-infection, including the representative members of *c1r* (complement component 1), *c2* (complement component 2), *c3* (complement component 3), *c7* (complement component 7), *c9* (complement component 9), *cfh* (complement factor h), *c1qbp* (complement component 1q subcomponent-binding protein, mitochondrial) and *c1qtnf6* (complement c1q tumor necrosis factor-related protein 6) (Figure 4). The expressions of *c1r*, *c2*, *c3*, *c7*, *c8b*, *c9*, *cfh* and *c1qtnf6* were up-regulated in the control group at 4 hpi and then decreased gradually at 24 hpi (*p* > 0.05), indicating their rapid participation in the host defense response of *M. amblycephala* against *A. hydrophila* infection. In addition, the expression of *c1r*, *c6*, *c1qbp* and *c1qtnf6* in MOS200 or MOS400 groups were higher than that of the control group before or post-infection, which revealed the activation of these genes by dietary MOS and might enhance the anti-infection immunity of *M. amblycephala* intestines.

### 2.5. Activation of Inflammation-Related Cytokines by MOS

A total of 18 inflammation-related DEGs were identified; these cytokines include *tnfaip3*, *tnf*, *tnfaip2*, *csf3r* and the like (Figure 5). The expression of *Ifnar1*, *Ifngr1*, *tnfaip2* and *tnfaip3* was significantly (*p* < 0.05) increased in the MOS200 or MOS400 groups post-*A. hydrophila* infection, especially at 4 hpi, while that of most DEGs in the control group showed no significant (*p* > 0.05) difference upon infection. These findings indicated that dietary MOS might enhance a host pro-inflammatory response by activating cytokines upon *A. hydrophila* infection.

### 2.6. Transcriptional Factors Activated by MOS

As shown in Figure 6, a total of 30 transcriptional regulation-related DEGs were identified. Most of them were significantly (*p* < 0.05) up-regulated in the MOS400 group at 4 hpi, while their expression increased slowly in the control group with partial up-regulation observed at 24 hpi, which revealed that these transcriptional factors could be activated more rapidly by dietary MOS. Specifically, the expression levels of *taf3* (transcription initiation factor TFIID subunit 3), *taf5* (transcription initiation factor TFIID subunit 5), *gtf2a1* (transcription initiation factor IIA subunit 1), *gtf2e1* (general transcription factor IIE subunit 1), *gtf2f2* (general transcription factor IIF subunit 2) and *ccnh* (Cyclin-H) were significantly (*p* < 0.05) up-regulated in dietary MOS groups, indicating that MOS might mediate the immune response of *M. amblycephala* via these transcriptional factors.

### 2.7. Apoptosis-Related Genes Regulated by MOS

As shown in Figure 7, part of the apoptotic proteases were down-regulated in the control group at 4 hpi, such as *caspase 1*, *caspase 2*, *caspase 3*, *caspase 7* and *caspase 10*, while anti-apoptotic factor *mcl1* was up-regulated (*p* > 0.05), revealing that intestinal apoptosis might occur rapidly post-infection in the control group and thereby lead to host feedback regulation. On the contrary, the expression of *caspase 7*, *caspase 8*, *caspase 9*, *bcl10* and *mcl1* was significantly (*p* < 0.05) decreased in the MOS groups at 24 hpi, and anti-apoptotic factor *bcl2* was significantly (*p* < 0.05) increased, which might be the protective pattern as the apoptosis mainly began to occur and was obvious at 24 hpi in the MOS groups. These data suggest that the MOS could effectively delay intestinal apoptosis and inhibit excessive apoptosis caused by the *A. hydrophila* infection in juvenile *M. amblycephala*.

### 2.8. Identification of Tight Junction-Related Genes

The tight junction-related genes contribute to the maintenance of intestinal barrier stabilization. As shown in Figure 8, most claudin (CLDN) family members (except for *claudin 4*) showed higher expression levels in the control group than that of the dietary MOS groups at 0 hpi, which were up-regulated in the MOS groups at 4 hpi and reached peak levels (*p* > 0.05), and then decreased at 24 hpi in all of the three groups. In addition, other tight junction-related genes showed different expression patterns upon infection. The expression of *PILT* (tight junction protein 4) significantly (*p* < 0.05) increased in the MOS groups upon infection and peaked at 4 or 24 hpi, while *ZO-2* (zonula occludens 2) and *occludin* showed a higher expression in the MOS groups at 0 hpi and then decreased gradually post-infection (*p* > 0.05). These results indicated that MOS supplementation induced the expression of tight junction-related genes and thereby maintained the intestinal tight junction structures upon infection.

To better understand the effects of dietary MOS and bacterial infection on the dynamic expression patterns of tight junction-related proteins and transcriptional regulators, western blot was performed and the protein levels of *M. amblycephala* claudin 3, occludin, ZO-1, p38, PKCα and NF-κB p65 in the intestines are shown in Figure 9. At the end of the culture experiment, the protein levels of p65, PKCα and ZO-1 in the dietary MOS groups were significantly (*p* < 0.05) higher than that in the control group, and most of the detected proteins were observably induced upon infection. The protein levels of tight junction-related proteins (ZO-1, occludin and claudin 3) reached their peak level at 4 hpi and then decreased, while those of the transcriptional regulators (p38, PKCα and p65) up-regulated and maintained high levels post-infection. These results also indicated that dietary MOS could maintain the intestinal tight junction structures.

### 2.9. qRT-PCR Verification

Thirty-four DEGs were randomly selected for real-time quantitative reverse transcription polymerase chain reaction (qRT-PCR) verification, including *ap5z1* (apolipoprotein a-I-2), *acss2* (acetyl-coenzyme a synthetase, cytoplasmic), *ahsg* (alpha-2-hs-glycoprotein), *alpl* (alkaline phosphatase, tissue-nonspecific isozyme), *akd1* (adenylate kinase domain-containing protein 1), *amy2* (pancreatic alpha-amylase), *batf* (basic leucine zipper transcriptional factor ATF-like), *c3* (complement c3), *crtc2* (CREB-regulated transcription coactivator 2), *cfh* (complement factor h), *cpa2* (carboxypeptidase a2), *ctrb1* (chymotrypsinogen 2), *cel* (bile salt-activated lipase), *c1q14* (complement c1q-like protein 4), *crys* (crystal protein), *cda* (cytidine deaminase), *fga* (fibrinogen alpha chain), *fgg* (fibrinogen gamma chain), *fgb* (fibrinogen beta chain), *f2* (prothrombin), *g4* (Ig heavy chain V-III region CAM), *hsp70* (heat shock 70 kDa protein), *hsp30* (heat shock protein 30), *itln1* (Intelectin-1), *lgmn* (legumain), *mgmt* (methylated-DNA--protein-cysteine methyltransferase), *nmt3* (phosphoethanolamine N-methyltransferase 3), *npr3* (atrial natriuretic peptide receptor 3), *prf1* (perforin-1), *pck1* (phosphoenolpyruvate carboxykinase, cytosolic), *rimbp2* (RIMS-binding protein 2), *sox21a* (transcription factor sox-21-a), *trpm5* (transient receptor potential cation channel subfamily m member 5) and *ydac* (uncharacterized methyltransferase ydac), and *GAPDH* was selected as the reference gene. The expression patterns of these genes were analyzed using qRT-PCR, which were compared with that of DGE profiling. As shown in Figure 10, all of these selected genes shared similar expression patterns between qRT-PCR and DGE profiling, revealing the reliability of DGE profiling.

## 3. Discussion

*M. amblycephala* is one of the major freshwater aquaculture species in China, while a gradual decline of disease resistance and the frequent occurrence of bacterial diseases are found during *M. amblycephala* aquaculture in recent years [20]. Bacterial septicemia caused by *A. hydrophila* infection is one of the most serious threats to the healthy aquaculture of *M. amblycephala* [17]. Our previous study has shown that MOS could effectively protect the intestinal tract of *M. amblycephala* and significantly reduce the morbidity challenged by *A. hydrophila* [11]. However, the potential molecular mechanism of the MOS on the intestinal barrier of *M. amblycephala* against the *A. hydrophila* infection is still lacking. Therefore, it is essential to study the dynamic gene expression patterns and regulatory mechanisms of the *M. amblycephala* immune system against the *A. hydrophila* infection.

In this study, comparative digital expression profiling was conducted to analyze the transcriptional regulation of dietary MOS and *A. hydrophila* infection on *M. amblycephala* intestines, so as to understand the underlying regulatory mechanisms. The expression levels of 34 randomly selected DEGs were verified by qRT-PCR; the results of *mgmt*, *fgb*, *ap5z1*, *f2*, *fgg*, *c3*, *itln1*, *alpl*, *sox1s*, *g4*, *gdac*, *akd1*, *amy2*, *ctrb1*, *cpa2*, *pck1*, *prf1*, *npr3*, *batf* and *crys* showed similar expression patterns as that of the DGE data (*p* < 0.05), indicating the reliability of the DGE profiling. Then, the regulatory mechanisms of MOS supplementation on the *M. amblycephala* intestinal immunity upon bacterial infection were predicted by GO enrichment and KEGG signaling pathway analysis of the identified DEGs [21,22], which revealed the affected biological functions of *M. amblycephala* intestines between different groups and time points.

According to the GO enrichment results, part of the DEGs enriched in the GO functional categories might relate to intestinal nutrients metabolism, barrier structures and biological functions, for instance, biological regulation, metabolic process and the response to stimulus of biological process, the cellular anatomical entity and intracellular of cellular component, the binding, catalytic activity, transcription regulator activity and molecular function regulator of molecular function. It could be speculated that MOS supplementation and *A. hydrophila* infection activated several signaling transduction pathways related to nutrients metabolism and cellular or biological processes, thus playing an important role in regulating the digestion and metabolism of nutrients, which was consistent with the improved feed utilization efficiency of juvenile *M. amblycephala* in the dietary MOS group [11].

Specific regulatory mechanisms that are involved in the complement, immune regulation, apoptosis, transcriptional regulation and tight junction of *M. amblycephala* intestines could be analyzed and verified based on the KEGG pathways enrichment. The identified DEGs of the MOS200 group vs. the control group were few at 4 hpi after infection, suggesting that MOS supplementation could reduce damage caused by excessive inflammatory responses in the intestine, thereby further maintaining host homeostasis. According to the KEGG pathways analysis of the DEGs, the most abundantly enriched pathways were mainly complement, immune, apoptosis and tight junction-related. These pathways should participate in the regulation of MOS supplementation and *A. hydrophila* infection to *M. amblycephala* intestines.

The complement system is an important component of the innate immune system and plays an important biological role in defense responses and immune regulation [23]. Complement is considered to be the first line of defense against pathogens because of its ability to rapidly mark and identify pathogens [24]. In this study, the expression of *c1r*, *c2*, *c3*, *c7*, *c8b*, *c9*, *cfh* and *c1qtnf6* were significantly up-regulated in the control group at 4 hpi post-infection, and the expressions of *c1r*, *c6*, *c1qbp* and *c1qtnf6* in the MOS200 or MOS400 groups were higher than that of the control group before or post-infection. It has been found that *ctrp6* exerts anti-inflammatory effects by activating extracellular-signal regulated protein kinase (ERK) 1/2, which can increase the expression of the anti-inflammatory factor *IL-10* of mononuclear macrophages in a dose-dependent manner [25]. Overexpression of CTRP6 decreases the expression of inflammatory factors *IL-1β*, *IL-6* and *TNF-α*, but increases the expression of the anti-inflammatory factor *IL-10* [26].

Pro-inflammatory cytokines and anti-inflammatory cytokines have been in a dynamic balance to maintain the stability of immune functioning [25]. The previous study revealed that dietary MOS could up-regulate the expression of intestinal anti-inflammatory cytokines and down-regulate that of pro-inflammatory cytokines in grass carp (*Ctenopharyngodon idella*) [27,28], thereby effectively inhibiting the development of intestinal inflammation [29], which is consistent with the present study. In this study, in comparison with the control group, the expression of intestinal anti-inflammatory cytokines increased and that of pro-inflammatory cytokines decreased in the MOS groups. However, the study on common carp (*Cyprinus Carpio*) found that MOS could significantly increase the expression of intestinal pro-inflammatory cytokines (e.g., *IL-8*) [30]. This different response may be related to the level of MOS supplementation, the size of the experimental fish and the infective concentration of pathogens. It was also found that the expression levels of *tnf-α*, *tnfaip2*, *tnfaip3*, *ifnar*, *ifnar2*, *sigirr*, *csf3r* and *csf2rb* in the *M. amblycephala* intestines were significantly increased in the dietary MOS groups at 4 hpi, while that of these genes were not significantly induced in the control group post-infection. These results suggest that appropriate levels of dietary MOS can promote the host immune response and activate cytokines to exert immunomodulatory effects in the early stages of intestinal inflammation in *M. amblycephala*.

The stability of internal environmental homeostasis depends on cell proliferation, differentiation and apoptosis. Apoptosis is an active process of cell death to better survive in its environment [29]. Apoptosis plays important roles not only in the normal growth and homeostasis of organisms but also in the pathogenesis of bacterial infections. However, excessive apoptosis could destroy the physical barrier of the intestine in fish. Caspases (a family of cysteine proteases) are central regulators of apoptosis [31,32]. Apoptosis induction is generally accompanied by the activation of multiple caspases, such as *caspase-9, -8* and *-3*. In this study, the expression levels of *caspase 3* and *caspase 10* were significantly decreased in the MOS groups at 4 hpi, while anti-apoptotic factor *bcl2* was significantly increased. *Bcl2* is a negative regulator of apoptosis and is capable of suppressing apoptosis in a variety of cell systems [33,34]. Our previous study also found that compared with the MOS-supplemented group, the microvilli and junctional complexes in the control group were significantly destroyed after infection, and showed pathological characteristics such as histological structure disorder, nuclear atypia, increased pinocytotic vesicles and partial necrocytosis [11]. These data suggest that MOS could effectively inhibit the excessive apoptosis of *M. amblycephala* intestines caused by the *A. hydrophila* infection.

Tight junctions are multifunctional complexes formed by the assembly of multiple transmembrane proteins and intracellular cytoplasmic proteins [35]. The main transmembrane proteins are claudins, occludin and junctional adhesion molecules (JAMs), and cytoplasmic proteins include occluding small band proteins (ZO) and buckling band proteins (cingulin). Our results displayed that MOS supplementation upregulated the expression of most of the tight junction proteins and downregulated OCLN expression. In our previous study, we also found that the expression of tight junction-related genes increased post-infection, and the MOS supplemental groups maintained gene expression at stable high levels after 12 hpi. Combined with the TEM results, the intestinal epithelial barriers of the MOS supplemental groups were well-protected upon infection and goblet cells were found in the MOS400 group [11]. These results indicated that MOS supplementation induced the expression of tight junction-related genes and thereby enhanced tight junctions in the fish intestine under *A. hydrophila* challenge.

## 4. Materials and Methods

### 4.1. Ethics Statement

This study was approved by the Animal Care and Use Committee of Jiangsu Ocean University (protocol no. 2020-37, approval date: 1 September 2019). All animal procedures were performed according to the guidelines for the Care and Use of Laboratory Animals in China.

### 4.2. Experiment Diets and Feeding Trial

According to our previous study [11], an isonitrogenous and isoenergy basal diet with fish, soybean, cottonseed and canola meal as protein sources was prepared based on the nutritional requirements of *M. amblycephala*. Appendix A shows the experimental diet formulation and proximate composition analysis with three different levels of MOS supplementation (0, 200 and 400 mg/kg). The diets were prepared and stored at 4 °C until feeding. Moisture was determined by the direct drying method (GB5009.3-2016), crude protein was determined by the automatic KjeltecTM-8400 (FOSS, Sonderborg, Denmark) and crude fat was determined by the SoxtecTM8000 Extraction Unit (FOSS, Sonderborg, Denmark).

Healthy juvenile *M. amblycephala* were obtained from a fish farm in Guangzhou, China. The experimental fish were sterilized for 10 minutes in 3% salt water before being placed in the tank for 2 weeks. After 2 weeks of temporary rearing and taming on a basal diet, fish husbandry was carried out in an indoor freshwater recirculating system comprised of 9 tanks (90 L/tank) with equal supplemental aeration and water flow (1 L/min). With an initial weight of 0.87 ± 0.05 g, 315 juvenile *M. amblycephala* were divided into three groups at random: control, MOS200 and MOS400. Each group had three replicates (35 fish per tank). The experimental fish were cultured for 8 weeks, and fed four times daily (8:00, 11:00, 14:00 and 17:00) to apparent satiation, with the water being renewed daily to maintain excellent water quality. The water was kept at a constant temperature of 26 to 28 °C, with a pH of about 7.2, ammonia, nitrogen and nitrite concentrations below 0.1 mg/L and dissolved oxygen levels over 6.0 mg/L.

### 4.3. Experimental Infections and Samples Collection

Following an 8-week culture, bacterial challenges were carried out as previously mentioned [36,37]. In total, 35 juvenile *M. amblycephala* from each tank were intraperitoneally injected with 0.1 mL of *A. hydrophila* at a concentration of 1 × 10^6^ CFU/mL (LD50 dose) at the end of the culture experiment. Three individuals from each tank were randomly dissected, and the midgut was extracted at 0, 4 and 24 hours after infection. The anesthesia was provided by 3-aminobenzoic acid ethyl ester methane sulfonate (MS-222; Merck KGaA, Darmstadt, Germany). For RNA extraction, all tissues were immediately frozen in liquid nitrogen and kept at −80 °C.

### 4.4. Total RNA Isolation, Illumina Sequencing and cDNA Preparation

The RNA Easy Fast Tissue Kit (TIANGEN, Beijing, China) was used to extract the total RNA from the intestines, which were then used for DGE profiling and cDNA preparation. Agarose gel electrophoresis and NanoDrop 2000 (Thermo Fisher Scientific, Wilmington, DE, USA) were used to assess the quality and quantity of the total RNA. BGI Genomics then used the total RNA to create sequencing libraries and carried out illumina sequencing for DGE profiling on a DNBSEQ platform (Wuhan, China). Furthermore, the cDNA was synthesized using the PrimeScript^®^ RT reagent kit with gDNA Eraser (TaKaRa, Dalian, China) in accordance with the manufacturer’s instructions and kept at −20 °C for qRT-PCR assay.

### 4.5. Sequencing Quality Control and Reads Mapping

By removing reads with low quality and joint contamination as well as an excessive amount of unknown base N content from the raw data of DGE profiling, clean reads were obtained. The clean reads that had been filtered were saved in a FASTQ format [38]. The sequencing data was filtered using SOAPnuke v1.5.2 (https://github.com/BGI-flexlab/SOAPnuke) (accessed on 2 December 2016). HISAT [39,40] was used to align the clean reads to the reference *M. amblycephala* genome, and Bowtie 2 [41] was used to calculate the gene alignment rate. The expression of unigenes was then calculated using RSEM [42], and the FPKM (expected number of fragments per kilobase of transcript sequence per millions of base pairs sequenced) of each unigene was calculated based on gene length and reads count mapping.

### 4.6. Differential Expression Analysis

Using the PossionDis algorithm [43], differential expression analysis was carried out between the three groups. Controlling the false discovery rate led to the determination of the field value of *p*-value (FDR) [44]. The differential expression multiples of unigenes from the three groups were simultaneously calculated based on the FPKM value to provide the difference test of FDR. The lower the FDR value, the larger the difference multiples and the greater the expression difference. In this study, the DEGs were determined to be FDR ≤ 0.05 and |log2(fold-change)| ≥ 1. In order to implement the GO enrichment and KEGG pathway analysis of DEGs, the R software’s phyper function was employed. Selected DEGs from several immune-related pathways were standardized using the Z-score approach, and heatmaps were created in R using the pheatmap package [45].

### 4.7. qRT-PCR Assay

As previously reported [28,46], 34 randomly chosen DEGs had their expression patterns examined using the qRT-PCR assay. Briefly, qRT-PCR was carried out using the QuantiNovaTM SYBR^®^ Green PCR Kit (TaKaRa) in accordance with the manufacturer’s instructions on an ABI StepOne Plus real-time PCR system (Applied Biosystems, California, USA). According to the analysis of geNorm software [47], *GAPDH* (glyceraldehyde-3-phosphate dehydrogenase) was chosen as the reference gene when comparing the relative expression levels of target genes using the 2^−∆∆Ct^ method [48]. The primers were presented in Table 4, and all reactions were carried out in triplicate. The relative expression levels of the MOS supplementary groups were displayed as fold changes, with the gene expression levels in the control group set to 1.

### 4.8. Western Blot 

Western blot was carried out as previously described in order to confirm the expression levels of numerous intestinal tight junction-associated proteins and signaling molecules. [49]. Using the Bradford Protein Assay Kit (Beyotime, Beijing, China), the total proteins of the *M. amblycephala* intestines were extracted and the concentrations were measured (Beyotime, Beijing, China). The protein samples were separated using 12% SDS-polyacrylamide gel electrophoresis (SDS-PAGE) gels and transferred to polyvinylidene difluoride (PVDF) membranes at 80 V for 2 h using a Trans-Blot instrument (BioRad, Hercules, CA, USA). Non-specific reactions were blocked with TBS (150 mM NaCl, 20 mM Tris-base, pH 7.4) containing 5% (*w*/*v*) skim milk powder for 1 h at 30 °C, followed by three washes with TBST for 10 min each. The primary antibodies at proper dilutions (Table 5) were treated with PVDF membranes for an overnight incubation at 4 °C. The SDS-PAGE analysis of the induced rMaOccludin protein is displayed in Appendix A, and the specificity analysis of the prepared MaOccludin antibody is displayed in Appendix A. Following incubation, the PVDF membranes were washed three times with TBST before being incubated for 1 h at 30 °C with horseradish peroxidase (HRP)-conjugated goat anti-rabbit/mouse IgG (H + L) (Beyotime; 1: 2000 dilution). Finally, the DAB Horseradish Peroxidase Color Development Kit (Beyotime) was used to visualize the bands. The target bands were calculated by the ImageJ system, and the grayscale values of the target bands were obtained for analysis. In this experiment, GAPDH was used as an internal reference for protein normalization, and the ratio of the grayscale value of the target protein to the grayscale value of the internal reference protein was used to express the relative expression of the target protein. The values were expressed as mean ± SE followed by Tukey’s HSD test, and *p* < 0.05 is statistically significant.

### 4.9. Statistical Analysis

The data in this study were presented as mean ± SE. SPSS 17.0 was used to perform a one-way analysis of variance (ANOVA) to determine the statistical significance. Tukey’s HSD was used to compare the statistical significance between t groups, and *p*-value < 0.05 was considered as a statistically significant difference.

## 5. Conclusions

In this study, the protective mechanisms of three different levels of dietary MOS supplementation on the intestinal immunity of juvenile *M. amblycephala* were investigated. GO and KEGG enrichment analyses of DEGs were performed to uncover the dynamic immune responses of juvenile *M. amblycephala* intestines at different time points post-infection. Here, we found the expression of complement-related DEGs was significantly up-regulated in the control group, indicating their rapid participation in the host defense response of *M. amblycephala* against *A. hydrophila* infection. The expression of these genes in the MOS-supplemented groups was higher than that of the control group before or post-infection, suggesting that these genes are activated by dietary MOS and may enhance the anti-infection immunity of *M. amblycephala* intestines. Furthermore, the up-regulation of DEGs associated with apoptosis suggested that MOS could effectively delay intestinal apoptosis and inhibit excessive apoptosis caused by the *A. hydrophila* infection in juvenile *M. amblycephala*. In addition, transcriptional factors, pro-inflammatory and tight junction-related genes were more significantly up-regulated upon infection in the dietary MOS groups to enhance host immune functions and maintain the stability of the intestinal barrier. In summary, these data will be helpful to clarify the regulatory mechanism of MOS supplementation on the intestinal barrier and immunity of juvenile *M. amblycephala*. These results also lay the theoretical foundation for the prevention and protection of fish bacterial diseases in the future.

## Figures and Tables

**Figure 1 ijms-24-02207-f001:**
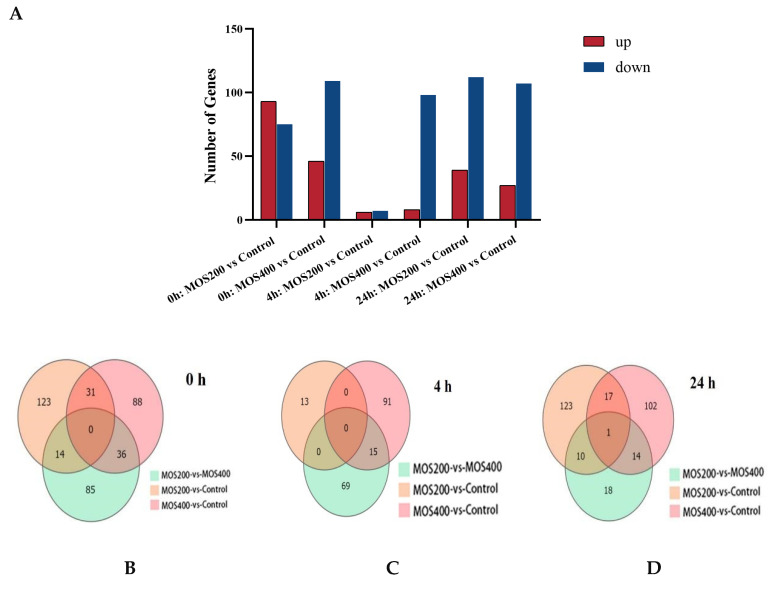
Statistics of differentially expressed genes (DEGs) at different time points upon infection. (**A**): Histogram analysis of DEGs. (**B**–**D**): Venn diagram analysis of DEGs at different time points.

**Figure 2 ijms-24-02207-f002:**
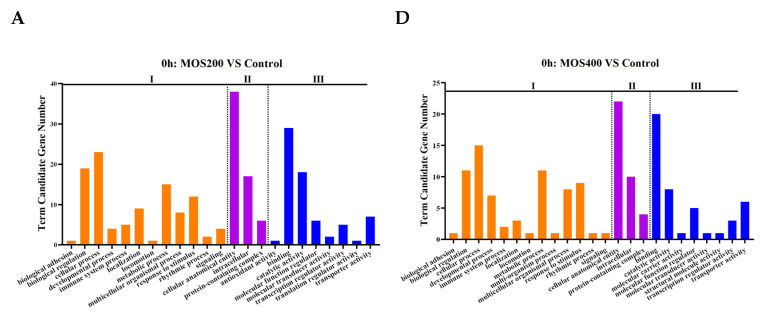
The mainly enriched Gene Ontology (GO) terms of identified DEGs. I: biological process, II: cellular component, III: molecular function. (**A**–**C**) was GO enrichment of DEGs between MOS200 and control groups at 0, 4 and 24 hpi, respectively. (**D**–**F**) was GO enrichment of DEGs between MOS400 and control groups at 0, 4 and 24 hpi, respectively.

**Figure 3 ijms-24-02207-f003:**
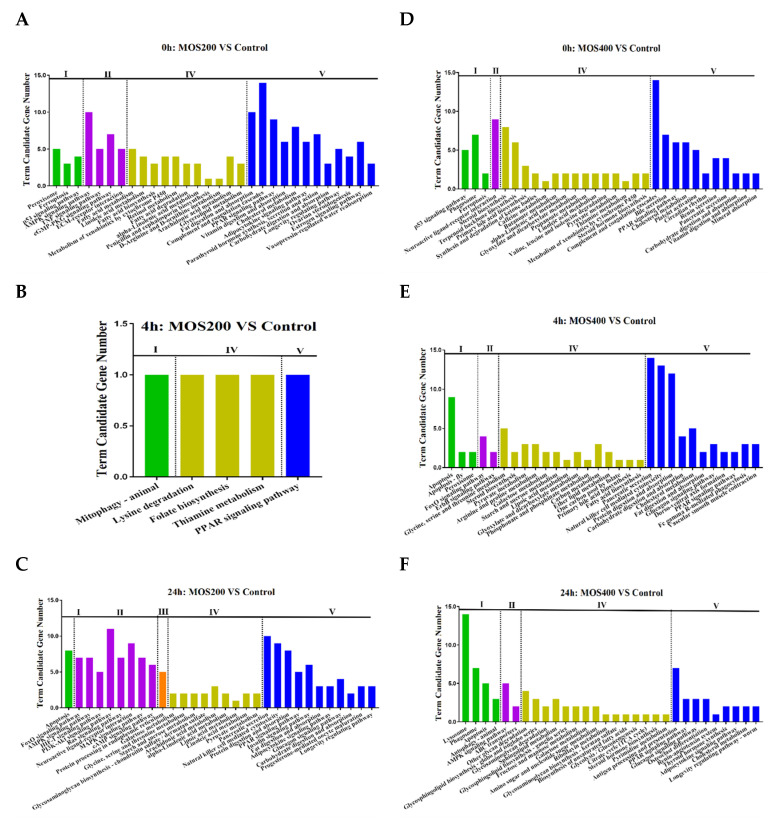
The 30 mainly enriched Kyoto Encyclopedia of Genes and Genomes (KEGG) terms of identified DEGs. I: cellular processes, II: environmental information processing, III: genetic information processing, IV: metabolism, V: organismal systems. **A**–**C** was KEGG enrichment of DEGs between MOS200 and control groups at 0, 4 and 24 hpi, respectively. **D**–**F** was KEGG enrichment of DEGs between MOS400 and control groups at 0, 4 and 24 hpi, respectively.

**Figure 4 ijms-24-02207-f004:**
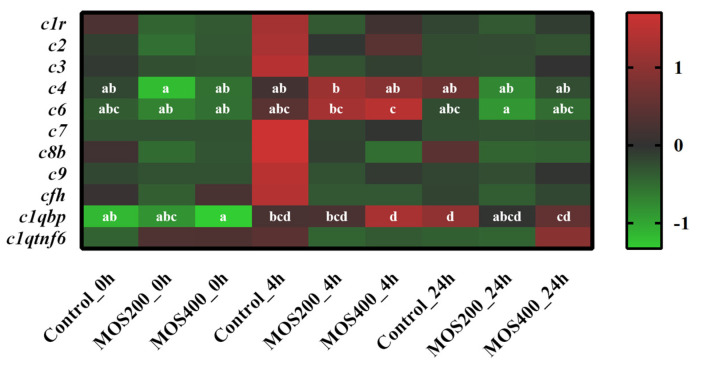
Heatmap of complement-related genes based on digital gene expression (DGE) profiling. The color values represented up-regulated (red) and down-regulated (green) expressions of selected DEGs, which ranged from 1 to −1. Different alphabets indicate statistically significant differences after performing Tukey’s HSD test (*p* < 0.05).

**Figure 5 ijms-24-02207-f005:**
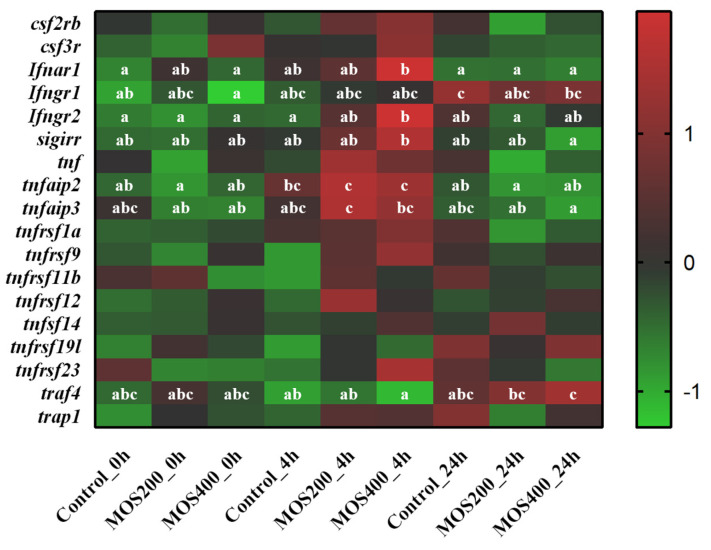
Heatmap of pro-inflammatory cytokines based on DGE profiling. The color values represented up-regulated (red) and down-regulated (green) expressions of selected DEGs, which ranged from 1 to −1. Different alphabets indicate statistically significant differences after performing Tukey’s HSD test (*p* < 0.05).

**Figure 6 ijms-24-02207-f006:**
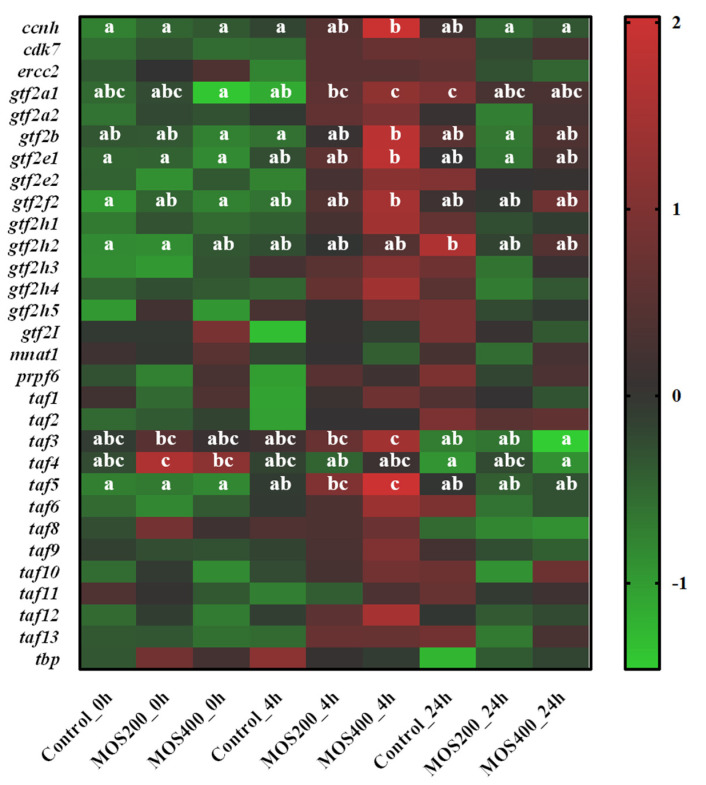
Heatmap of transcriptional regulation-related DEGs based on DGE profiling. The color values represented up-regulated (red) and down-regulated (green) expressions of selected DEGs, which ranged from 2 to −1. Different alphabets indicate statistically significant differences after performing Tukey’s HSD test (*p* < 0.05).

**Figure 7 ijms-24-02207-f007:**
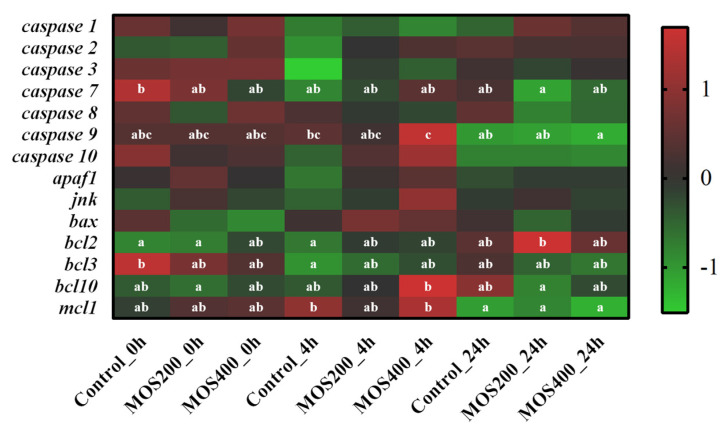
Heatmap of apoptosis-related genes based on DGE profiling. The color values represented up-regulated (red) and down-regulated (green) expressions of selected DEGs, which ranged from 1 to −1. Different alphabets indicate statistically significant differences after performing Tukey’s HSD test (*p* < 0.05).

**Figure 8 ijms-24-02207-f008:**
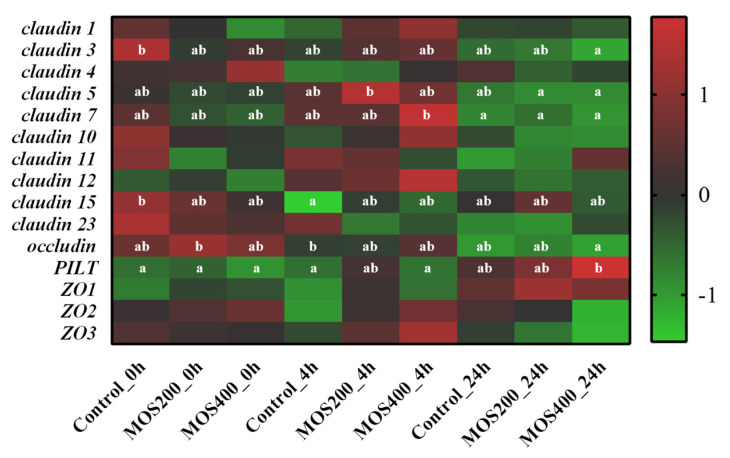
Heatmap of tight junction-related genes based on DGE profiling. The color values represented up-regulated (red) and down-regulated (blue) expressions of selected DEGs, which were ranged from 1 to −1. Different alphabets indicate statistically significant differences after performing Tukey’s HSD test (*p* < 0.05).

**Figure 9 ijms-24-02207-f009:**
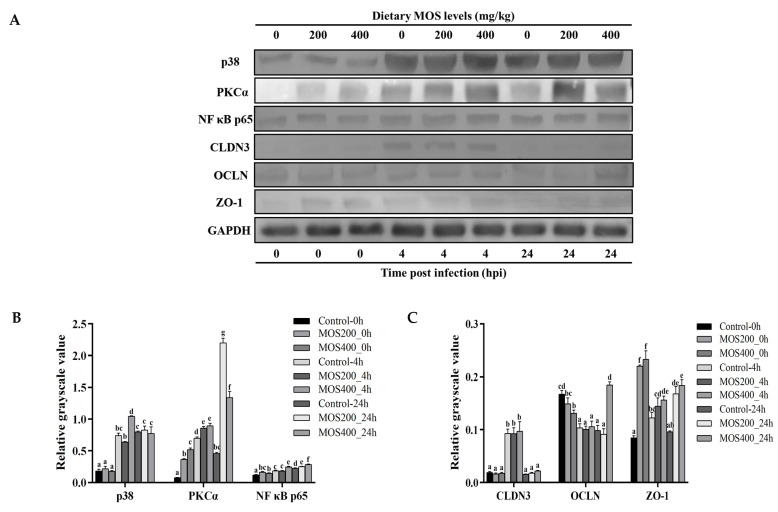
The expression of partial tight junction-related proteins and transcriptional regulators in the intestines of *M. amblycephala* was analyzed by western blot. (**A**) Western blot analyzed the expression of partial tight junction-related proteins and transcriptional regulators. The relative expression of tight junction-related proteins (**B**) and transcriptional regulators (**C**) was calculated by the gray value of the western blot. Different alphabets indicate statistically significant differences after performing Tukey’s HSD test (*p* < 0.05).

**Figure 10 ijms-24-02207-f010:**
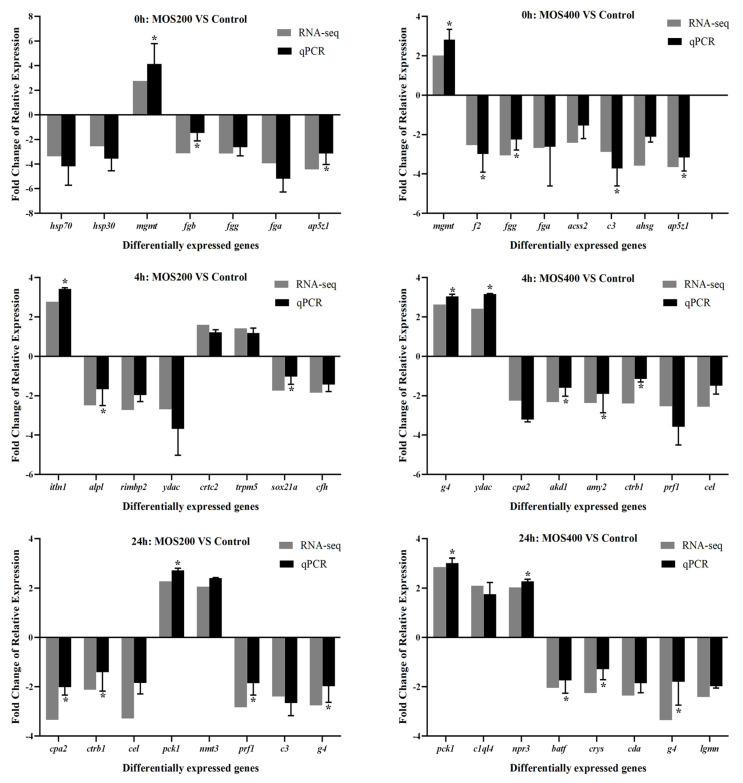
qRT-PCR validation of the gene expression patterns of randomly selected DEGs in the intestines of dietary MOS and control groups at different time points post-infection, and *GAPDH* was selected as the internal reference gene. Data were shown as mean ± SE. Asterisks indicate statistically significant differences after performing Tukey’s HSD test (*p* < 0.05).

**Table 1 ijms-24-02207-t001:** Data quality of digital gene expression (DGE) profiling.

Samples	Raw Reads (M)	Clean Reads (M)	Error Rate (%)	Q20 (%)	Q30 (%)	GC Content (%)
Control_0_1	21.75	21.42	0.03	97.90	93.55	45.34
Control_0_2	21.75	21.47	0.03	97.84	93.38	46.73
Control_0_3	21.75	21.49	0.02	98.03	93.92	47.11
Control_4_1	21.75	21.50	0.02	98.06	93.99	46.51
Control_4_2	21.75	21.48	0.03	98.08	94.11	46.45
Control_4_3	21.75	21.48	0.03	98.33	94.47	47.33
Control_24_1	21.75	21.49	0.02	98.13	94.28	45.93
Control_24_2	21.75	21.5	0.03	97.88	93.54	46.14
Control_24_3	21.75	21.46	0.02	98.03	94.01	46.13
MOS200_0_1	20.44	20.15	0.03	97.82	93.55	46.38
MOS200_0_2	21.75	21.37	0.03	97.85	93.57	46.03
MOS200_0_3	21.75	21.46	0.03	98.02	94.06	45.56
MOS200_4_1	21.75	21.48	0.03	98.00	93.85	46.26
MOS200_4_2	21.75	21.45	0.02	97.75	93.27	46.67
MOS200_4_3	21.75	21.45	0.02	97.94	93.87	45.79
MOS200_24_1	21.75	21.46	0.03	97.99	93.84	46.02
MOS200_24_2	21.75	21.47	0.03	98.26	94.6	45.80
MOS200_24_3	21.75	21.46	0.03	98.20	94.38	45.85
MOS400_0_1	20.35	20.07	0.03	97.76	93.43	46.45
MOS400_0_2	21.75	21.44	0.03	98.06	94.10	45.73
MOS400_0_3	21.75	21.48	0.03	98.07	94.13	45.67
MOS400_4_1	21.75	21.43	0.03	97.97	93.85	46.33
MOS400_4_2	21.75	21.45	0.02	97.81	93.53	46.14
MOS400_4_3	21.75	21.48	0.03	98.01	93.99	46.21
MOS400_24_1	21.75	21.48	0.03	98.06	94.04	46.47
MOS400_24_2	21.75	21.46	0.04	98.02	93.86	46.29
MOS400_24_3	21.75	21.48	0.03	97.90	93.64	45.84

**Table 2 ijms-24-02207-t002:** Data quality of DGE profiling.

Groups	GO (%)	KEGG (%)	Up-Regulation (%)	Down-Regulation (%)	Total
0 h MOS200 vs. Control	53 (31.55)	155 (92.26)	93 (55.36)	75 (44.64)	168
0 h MOS400 vs. Control	38 (24.52)	150 (96.77)	46 (29.68)	109 (70.32)	155
4 h MOS200 vs. Control	3 (23.08)	9 (69.23)	6 (46.15)	7 (53.85)	13
4 h MOS400 vs. Control	35 (33.02)	101 (95.28)	8 (7.55)	98 (72.45)	106
24 h MOS200 vs. Control	54 (35.76)	132 (87.42)	39 (25.83)	112 (74.17)	151
24 h MOS400 vs. Control	33 (24.63)	126 (94.03)	27 (20.15)	107 (79.85)	134
Total	176	570	198	434	622

**Table 3 ijms-24-02207-t003:** The top 10 enriched Kyoto Encyclopedia of Genes and Genomes (KEGG) pathways of identified differentially expressed genes (DEGs) at different time points post-infection.

Ko ID	KEGG Terms	Number of DEGs	FDR ≤ 0.01
**0 h MOS200 vs. Control**			
map04975	Fat digestion and absorption	10	5.01 × 10^−10^
map04610	Complement and coagulation cascades	14	8.37 × 10^−9^
map03320	PPAR signaling pathway	9	1.98 × 10^−6^
map04977	Vitamin digestion and absorption	6	2.45 × 10^−5^
map04979	Cholesterol metabolism	8	2.15 × 10^−5^
map04152	AMPK signaling pathway	10	4.47 × 10^−5^
map04920	Adipocytokine signaling pathway	6	1.31 × 10^−3^
map01212	Fatty acid metabolism	5	2.16 × 10^−3^
map00591	Linoleic acid metabolism	4	2.91 × 10^−3^
map00061	Fatty acid biosynthesis	3	3.47 × 10^−3^
**0 h MOS400 vs. Control**			
map00100	Steroid biosynthesis	8	3.00 × 10^−11^
map04610	Complement and coagulation cascades	14	1.88 × 10^−9^
map00900	Terpenoid backbone biosynthesis	6	2.10 × 10^−7^
map04976	Bile secretion	7	8.21 × 10^−5^
map03320	PPAR signaling pathway	6	5.40 × 10^−4^
map04979	Cholesterol metabolism	6	6.16 × 10^−4^
map00120	Primary bile acid biosynthesis	3	1.54 × 10^−3^
map04115	p53 signaling pathway	5	5.83 × 10^−3^
**4 h MOS200 vs. Control**			
map00730	Thiamine metabolism	1	5.42 × 10^−3^
**4 h MOS400 vs. Control**			
map04972	Pancreatic secretion	14	1.17 × 10^−12^
map04650	Natural killer cell-mediated cytotoxicity	13	1.47 × 10^−9^
map04974	Protein digestion and absorption	12	1.52 × 10^−8^
map00260	Glycine, serine and threonine metabolism	5	7.61 × 10^−6^
map04210	Apoptosis	9	4.12 × 10^−5^
map04973	Carbohydrate digestion and absorption	4	2.67 × 10^−4^
map04979	Cholesterol metabolism	5	5.49 × 10^−4^
map00100	Steroid biosynthesis	2	8.93 × 10^−3^
map00620	Pyruvate metabolism	3	8.44 × 10^−3^
**24 h MOS200 vs. Control**			
map04972	Pancreatic secretion	10	4.97 × 10^−7^
map04650	Natural killer cell-mediated cytotoxicity	9	6.92 × 10^−5^
map04974	Protein digestion and absorption	8	3.78 × 10^−4^
map04068	FoxO signaling pathway	7	6.60 × 10^−4^
map04152	AMPK signaling pathway	7	6.78 × 10^−4^
map04210	Apoptosis	8	1.05 × 10^−3^
map03320	PPAR signaling pathway	5	1.26 × 10^−3^
map04066	HIF-1 signaling pathway	5	8.01 × 10^−3^
**24 h MOS400 vs. Control**			
map04910	Insulin signaling pathway	6	9.62 × 10^−3^
map04142	Lysosome	14	3.62 × 10^−12^
map00511	Other glycan degradation	4	5.29 × 10^−5^
map04612	Antigen processing and presentation	7	8.44 × 10^−5^
map00603	Glycosphingolipid biosynthesis—globo and isoglobo series	3	2.71 × 10^−4^
map04152	AMPK signaling pathway	5	6.07 × 10^−3^
map00531	Glycosaminoglycan degradation	2	9.05 × 10^−3^
map04145	Phagosome	7	7.95 × 10^−3^

**Table 4 ijms-24-02207-t004:** Primers used for qRT-PCR in the present study.

Target Genes	Primer Sequences (5′-3′)
*ap5z1*	Forward: GCTGGATCATGTGAAGTCT Reverse: CACCTGGGTAGCAACTGG
*acss2*	Forward: GACAGAGTCGCCATCTACCT Reverse: CAGAGCCTCAGACGCAAT
*ahsg*	Forward: ACACCCACGGTTACAAAT Reverse: AATGCCACAACAGACAGAC
*alpl*	Forward: ACTCCCACGCATTTACCT Reverse: GTCTGGACGCTTGTTGTTA
*ak9*	Forward: ACACTGAAACCGACTGAA Reverse: CACTAAAGGCATCCATCT
*amy2*	Forward: CCGACTACATGAACAAGC Reverse: CTCAGTCACCCTTCCAAT
*batf*	Forward: AAAAGAAATCGCTGAACTG Reverse: GAATCTGAGGTAAAGGGTG
*c3*	Forward: ACAAGCCCATCTACACGC Reverse: ATCCCAGAACTGGCAACC
*crtc2*	Forward: GGTTCTTGTCCCGCTCCC Reverse: TCTCCTCCGCCTGCCTCT
*cfh*	Forward: AAAGATAGACGGAAGTAGTGA Reverse: TTGAACCCTGGTGAACAT
*cpa2*	Forward: CATTGCGGACTTCATCAC Reverse: AGGGCTTTGTTTGCTTGTG
*ctrb1*	Forward: CACTTCTGCGGTGGCTCT Reverse: CAGGAGGATGTCGTTGTTGA
*cel*	Forward: GGAAAGGTTATCGTGGTG Reverse: GGAAGTTTACGCTGGCTC
*c1q14*	Forward: TGTGGTCCAGAAGGTAGA Reverse: CTCAGTAGCCCTCAGTCG
*cryS*	Forward: CGAAACCAGGACAGAAGA Reverse: TTGTAGCAGAACGGAAGC
*cda*	Forward: TACTGTCCCTACAGCAAAT Reverse: TGGAGAAATAAACCGATC
*fga*	Forward: CAACAGGGTAAGGGCGAGAT Reverse: TCTGCCTCCGAGCCAACT
*fgg*	Forward: CGTGCCTTATGTGCTGAG Reverse: GTGCGAACTTGTCATTGTCT
*fgb*	Forward: TACATCCTAGCAGTTGGTCGTT Reverse: CTCCAGGTGTCCATTTGTCATT
*f2*	Forward: CCTTCACGACCTCCTACT Reverse: ACCCTCCATATCTCCATCT
*g4*	Forward: CTGGAGAATCCCATAAAC Reverse: CAGATACACCTGCTTCCT
*hsp70*	Forward: ATGTGGCTCCTCTGTCCC Reverse: CAGCAGGTTGTTGTCTTTAGTC
*hsp30*	Forward: GGACGACCCTTTCTTTGC Reverse: GCCGCTTGTTCTCATCTGTG
*itln1*	Forward: TAGTGAGCAGGGCAGCAA Reverse: CAGGATGGAGGCAATGGT
*lgmn*	Forward: ACCTACCTGGGAGACTGG Reverse: CTTTAGAGTTACCCTGGAAT
*mmgmt*	Forward: GCGTGATGTGACTCTTTG Reverse: AACTCAGGGCTCATTTCT
*nmt3*	Forward: GGATAACCAGCAGTACACCC Reverse: TCCCACAGCCGACATCTA
*npr3*	Forward: GTCTACGGGAGCGAAGTG Reverse: TTTGGTGGACCTAAGTAATG
*prf1*	Forward: CTGCTGTTCGGGTGTTTG Reverse: CTCCAGCTTCAGGTTATCATTC
*pck1*	Forward: GTGCGCTCATCCCAACTC Reverse: ACACTCCGTGCGACCAAT
*rimp2*	Forward: GTCACCAAACCCTGAAAC Reverse: CTGCTGGAACAGACGAAG
*sox21a*	Forward: GAAGCCAAACGACTTAGAGC Reverse: CGAGTAGCAGCAGCAACAG
*trpm5*	Forward: TGGATTGCGGAGTGTTTC Reverse: TTGGCTTCCTGTGGTTGT
*ydaC*	Forward: CATACAACCCAACAACACG Reverse: CCTTCATGCGCTCACTAG
*GAPDH*	Forward: TGCCGGCATCTCCCTCAA Reverse: TCAGCAACACGGTGGCTGTAG

**Table 5 ijms-24-02207-t005:** The information of antibodies used for Western blot.

Antibodies	Host	Source	Catalog No.	Dilution for WB
p38	Rabbit	Abcam	ab254122	1: 1000
PKCα	Rabbit	GeneTex	GTX130453	1: 1000
NF-κB p65	Rabbit	Affinity	AF5006	1: 1000
Claudin-3	Rabbit	GeneTex	GTX135322	1: 1000
Occludin	Rabbit	Self-made	-	1: 500
ZO-1	Mouse	Thermo	33-9100	1: 500
GAPDH	Rabbit	GeneTex	GTX82899	1: 3000

## Data Availability

The raw sequencing data are available in the NCBI database (accession no. PRJNA846987).

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
