# Peer review of "Comparative Expression Profiling Reveals the Regulatory Effects of Dietary Mannan Oligosaccharides on the Intestinal Immune Response of Juvenile Megalobrama amblycephala against Aeromonas hydrophila Infection"

_ijms, 2023, doi:10.3390/ijms24032207_

Round 1
Reviewer 1 Report
The work emphasizes improvement in studies related to use of functional MOS towards fish immunity through (DGE) profiling to provide a holistic understanding of the gene expression and signaling pathways upon bacterial infection. Though quite lots of work have been accomplished in dietary prebiotic, the study is important as it covers a new framework of regulatory mechanism in fishes. The introduction is well narrated highlighting earlier cases, few recent work by Singh et al (2019) on FOS can be included. The work is meaningful, however few critical points for improving the work are mentioned below:
4.2. Experiment diets and feeding trial
The diet formulation is not available. Level of protein used should be mentioned. How was the MOS supplemented? How was the water quality parameter estimated? Whether disinfection prior to stocking was performed?
4.3. Experimental Infections and samples collection
Was the A. hydrophila was infective strain? If strain name available, please provide.
Which part of the intestine was taken as sample?
Conclusion section is poor. Please rewrite throwing light on the key finding and future course.
Author Response
Point 1: Moderate English changes required.
Response 1: Thank you for your comments. We have carefully revised the manuscript, and the grammatical or minor spell problems have been checked and corrected.
Point 2: The work emphasizes improvement in studies related to use of functional MOS towards fish immunity through (DGE) profiling to provide a holistic understanding of the gene expression and signaling pathways upon bacterial infection. Though quite lots of work have been accomplished in dietary prebiotic, the study is important as it covers a new framework of regulatory mechanism in fishes. The introduction is well narrated highlighting earlier cases, few recent work by Singh et al (2019) on FOS can be included. The work is meaningful, however few critical points for improving the work are mentioned below:
Response 2: Thank you for your comments. We have addressed all the concerns you have mentioned.
Point 3: 4.2. Experiment diets and feeding trial
The diet formulation is not available. Level of protein used should be mentioned. How was the MOS supplemented? How was the water quality parameter estimated? Whether disinfection prior to stocking was performed?
Response 3: Thank you for your questions. The diet formulation was described in our previous study, so we displayed the experimental diets formulation and proximate composition analysis in Supplementary Table S1. Isonitrogenous and isoenergy basal diet was prepared based on the nutritional requirements of M. amblycephala and the crude protein was determined by automatic KjeltecTM-8400 (FOSS, Denmark). MOS was used to replace 0.02% and 0.04% of Ca(H2PO4)2 in the basal diet, respectively. We measure the temperature and dissolved oxygen in the culture water every day, and test the water quality with a water quality tester every week. The experimental fish were sterilized by soaking in 3% salt water for 10 minutes and then put into the tank for two weeks temporarily. We have added these information in lines 349-364.
Point 4: 4.3. Experimental Infections and samples collection
Was the A. hydrophila was infective strain? If strain name available, please provide.
Response 4: Thank you for your questions. The A. hydrophila used in the present study was isolated from diseased M. amblycephala previously and was shown to be pathogenicity by artificial infection. Bacterial challenges were performed according to the our previously published articles that using the same A. hydrophila strain (Ding et al., 2017; Cui et al., 2022).
Ding, Z.; Zhao, X.; Zhan, Q.; Cui, L.; Sun, Q.; Lin, L.; Wang, W.; Liu, H. Characterization and expression analysis of an intelectin gene from Megalobrama amblycephala with excellent bacterial binding and agglutination activity. Fish Shellfish Immun. 2017, 61, 100-110. https://doi.org/10.1016/j.fsi.2016.12.023.
Cui, H.; Li, H.; Zhang, M.; Li, H.; Wang, X.; Wang, Z.; Zhai, W.; Chen, X.; Cheng, H.; Xu, J.; Zhao, X.; Ding, Z. Molecular characterization, expression, evolutionary selection, and biological activity analysis of CD68 gene from Megalobrama ambly-cephala. Int. J. Mol. Sci. 2022, 23 (21), 13133. https://doi.org/10.3390/ijms232113133.
Point 5: Which part of the intestine was taken as sample?
Response 5: Thank you for your question. In this study, we sampled the midgut of juvenile M. amblycephala, which have been added in line 369.
Point 6: Conclusion section is poor. Please rewrite throwing light on the key finding and future course.
Response 6: Thank you for your suggestions. We have revised the conclusion according to your suggestions.

Reviewer 2 Report
The reviewed manuscript is focused on the dietary application of complex carbohydrate molecule MOS for immuno-stimulation of M. amblycephala, stability of intestinal barrier against A. hydrophila. The manuscript highlighted the differential gene expression and digital gene expression profiling of the bacteria infected fish in a temporal manner. The authors nicely explain the underlying mechanism, associated GO and KEGG pathways of the outcome results. However you should consider expanding each of the components that you included in this manuscript.
Positive comments to the authors
Please have a look at my comments included in the attached file. I really liked the fact that you attempted to consider many aspects and that you have developed a good ground for many good manuscripts in the future. You have conducted a good study with huge relevance to the aquaculture field.
Constructive comments to the authors
The authors made effort to incorporate a number of interesting parameters. However I have a number of issues I have highlighted in the document attached, before the manuscript can qualify to be published in this Journal. Major revision is required.
Mainly the authors have to rewrite the manuscript giving emphasis on the statistical methodology and have to elaborately mention the significance value everywhere in the result section like "significantly (P<0.05) increased / not significantly (P>0.05) increased". Along with that they have to rebuild the figures.

Author Response
Point 1: English language and style are fine/minor spell check required.
Response 1: Thank you for your comments. We have carefully revised the manuscript, and the grammatical or minor spell problems have been checked and corrected.
Point 2: The reviewed manuscript is focused on the dietary application of complex carbohydrate molecule MOS for immuno-stimulation of M. amblycephala, stability of intestinal barrier against A. hydrophila. The manuscript highlighted the differential gene expression and digital gene expression profiling of the bacteria infected fish in a temporal manner. The authors nicely explain the underlying mechanism, associated GO and KEGG pathways of the outcome results. However you should consider expanding each of the components that you included in this manuscript.
Response 2: Thank you for your comments. We have revised the manuscript according to your suggestions.
Point 3: Please have a look at my comments included in the attached file. I really liked the fact that you attempted to consider many aspects and that you have developed a good ground for many good manuscripts in the future. You have conducted a good study with huge relevance to the aquaculture field.
Response 3: Thank you for your comments. We have made revisions according to the comments included in the attached file.
Point 4: The authors made effort to incorporate a number of interesting parameters. However I have a number of issues I have highlighted in the document attached, before the manuscript can qualify to be published in this Journal. Major revision is required.
Response 4: Thank you for your comments. We have made revisions according to the comments included in the attached file.
Point 5: Mainly the authors have to rewrite the manuscript giving emphasis on the statistical methodology and have to elaborately mention the significance value everywhere in the result section like "significantly (P < 0.05) increased / not significantly (P > 0.05) increased". Along with that they have to rebuild the figures.
Response 5: Thank you for your suggestions. We have performed Tukey's HSD after performing ANOVA to compare the means that are significantly (P < 0.05) different from each other. We have revised the result section and have rebuilt the Figure 4, Figure 5, Figure 6, Figure 7 and Figure 8 in the manuscript.
Point 6: Rewrite the abstract following the word limits mentioned in the author's guidelines.
Response 6: Thank you for your suggestions. We have revised the abstract following the word limits mentioned in the author's guidelines.
Point 7: On line 44, remove the sentence “With the expansion and promotion of the intensive aquaculture model, aquaculture has developed rapidly, but it has also brought many problems, such as the degradation of germplasm resources, water pollution and the frequent occurrence of aquatic animal diseases”.
Response 7: Thank you. We have removed the sentence.
Point 8: On line 54, add a reference to this sentence.
Response 8: Thank you for you suggestion. We have added a reference to this sentence.
Xu, T.; Sun, R.; Zhang, Y.; Zhang, C.; Wang, Y.; Wang, Z.; Du, Y. Recent research and application prospect of functional oligosaccharides on Intestinal disease treatment. Molecules. 2022, 27 (21), 7622. https://doi.org/10.3390/molecules27217622.
Point 9: On line 58, add a reference to this sentence.
Response 9: Thank you for you suggestion. We have added a reference to this sentence.
Liu, X.; Li, X.; Bai, Y.; Zhou, X.; Chen, L.; Qiu, C.; Lu, C.; Jin, Z.; Long, J.; Xie, Z. Natural antimicrobial oligosaccharides in the food industry. Int. J. Food. Microbiol. 2022, 386, 110021. https://doi.org/10.1016/j.ijfoodmicro.2022.110021.
Point 10: On line 118, add a reference to this sentence.
Response 10: Thank you for you suggestion. We have added two references to this sentence.
Hwang, J.; Kwon, M.; Jung, S.; Park, M.; Kim, D.; Cho, W.; Park, J.; Son, M. RNA-Seq transcriptome analysis of the olive flounder (Paralichthys olivaceus) kidney response to vaccination with heat-inactivated viral hemorrhagic septi-cemia virus. Fish. Shellfish. Immun. 2017, 62, 221-226. https://doi.org/10.1016/j.fsi.2017.01.016.
Wu Q.; Ning, X.; Jiang, S.; Sun, L. Transcriptome analysis reveals seven key immune pathways of Japanese flounder (Paralichthys olivaceus) involved in megalocytivirus infection. Fish. Shellfish. Immun. 2020, 103, 150-158. https://doi.org/10.1016/j.fsi.2020.05.011.
Point 11: On line 203, explain the reason behind this result in the discussion section
Response 11: Thank you. We have explain the reason in lines 628-630.
Point 12: On line 242, replace dramatically with significantly.
Response 12: Thank you for you suggestion. We have replaced the word.
Point 13: On line 276, replace markedly with significantly.
Response 13: Thank you for you suggestion. We have replaced the word.
Point 14: On line 287, remove the word “much”.
Response 14: Thank you for you suggestion. We have removed the word.
Point 15: On line 292, replace dramatically with significantly.
Response 15: Thank you for you suggestion. We have replaced the word.
Point 16: On line 297, replace understanding with understand.
Response 16: Thank you for you suggestion. We have replaced the word.
Point 17: On line 300, replace Figure 8 with Figure 9.
Response 17: Thank you for you suggestion. We have replaced the word.
Point 18: On line 311, with this figure authors are requested to attach a densitometric graphical representation of the proteins.
Response 18: Thank you for your suggestion. We have attached a grayscale value of the proteins (Fig. 9B and 9C). The relative expression of tight junction related proteins and signaling factors was calculated by the ImageJ software, and the grayscale values of the target bands were obtained for analysis.
Point 19: On line 335, replace Figure 9 with Figure 10.
Response 19: Thank you for your suggestion. We have replaced the word.
Point 20: On line 388, add a reference to this sentence.
Response 20: Thank you for your suggestion. We have added a reference to this sentence.
Kim, M.; Lee, W.; Park, E.; Park, S. C1qTNF-related protein-6 increases the expression of interleukin-10 in macrophages. Mol. Cells. 2010, 30, 59-64. https://doi.org/10.1007/s10059-010-0088-x.
Point 21: On line 418, replace markedly with significantly.
Response 21: Thank you for your suggestion. We have replaced the word.
Point 22: On line 524, write about the measurement of individual band intensity measurement, densitometry and graph preparation.
Response 22: Thank you for your suggestion. The target bands were calculated by the ImageJ software, and the grayscale values of the target bands were measured for analysis. In this experiment, GAPDH was used as an internal reference for protein normalization, and the ratio of the grayscale value of the target protein to the grayscale value of the internal reference protein was used to express the relative expression level of the target protein. We have added these description in lines 978-981, and added the grayscale images of Fig. 9B and 9C.
Point 23: On line 531, the authors have to perform a Honestly Significance Difference Test (HSD) like Tukey's HSD after performing ANOVA to compare the means that are significantly (P<0.05) different from each other. After performing the HSD test, include the result of difference in each graph used in the manuscript. I think a power analysis must be included in the results to make it strong in terms of sampling.
Response 23: Thank you for your suggestions. We have performed Tukey's HSD after performing ANOVA to compare the means that are significantly (P < 0.05) different from each other. We have rebuilt the Figure 4, Figure 5, Figure 6, Figure 7 and Figure 8 in the manuscript.
Point 24: On line 564, try to replace with a recent article.
Response 24: Thank you for your suggestion. We have replaced it with a recent article.
Zhang, N.; Jin, M.; Wang, K.; Zhang, Z.; Shah, N.; Wei, H. Functional oligosaccharide fermentation in the gut: Improving intestinal health and its determinant factors-A review. Carbohyd. Polym. 2021, 284 119043. https://doi.org/10.1016/j.carbpol.2021.119043.
Point 25: On line 598, try to replace with a recent article.
Response 25: Thank you for your suggestion. We have replaced it with a recent article.
Wang, T.; Wu, H.; Li, W.; Xu, R.; Qiao, F.; Du, Z.; Zhang, M. Effects of dietary mannan oligosaccharides (MOS) supple-mentation on metabolism, inflammatory response and gut microbiota of juvenile Nile tilapia (Oreochromis niloticus) fed with high carbohydrate diet. Fish Shellfish Immun. 2022, 130, 550-559. https://doi.org/10.1016/j.fsi.2022.09.052.
Point 26: On line 638, try to replace with a recent article.
Response 26: Thank you for your suggestion. We have replaced it with a recent article.
Li, H.; Wang, H.; Zhang, J.; Liu, R.; Zhao, H.; Shan, S.; Yang, G. Identification of three inflammatory Caspases in common carp (Cyprinus carpio L.) and its role in immune response against bacterial infection. Fish Shellfish Immun. 2022, 131, 590-601. https://doi.org/10.1016/j.fsi.2022.10.035.
Point 27: On line 669, try to replace with a recent article.
Response 27: Thank you for your suggestion. We have replaced it with a recent article.
Lynch, G.; Guo, W.; Sarkar, S.; Finner, H. The control of the false discovery rate in fixed sequence multiple testing. Electron. J. Stat. 2017, 11 (2), 4649-4673. https://doi.org/10.1214/17-EJS1359.

Round 2
Reviewer 2 Report
Please have a look at my comments included in the attached file. Minor revisions required.

Author Response
Point 1: English language and style are fine/minor spell check required.
Response 1: Thank you for your comments. We have carefully revised the manuscript, and the grammatical or minor spell problems have been checked and corrected.
Point 2: Please have a look at my comments included in the attached file. Minor revisions required.
Response 2: Thank you for your comments. We have made revisions according to the comments included in the attached file.
Point 3: On line 176, confirm whether it is cfh or cgh?
Response 3: Thank you for your comments. We have corrected it.
Point 4: On line 186, replace “superscript letters” with “alphabets”.
Response 4: Thank you for your comments. We have replaced the word.
Point 5: On line 187, add “after performing Tukey’s HSD test” after “differences”.
Response 5: Thank you for your suggestions. We have added the words.
Point 6: On line 190, add “(P < 0.05)” after “significantly”.
Response 6: Thank you for your suggestions. We have added the words.
Point 7: On line 192, add “(P > 0.05)” after “show no significant”.
Response 7: Thank you for your suggestions. We have added the words.
Point 8: On line 196, replace “superscript letters” with “alphabets”.
Response 8: Thank you for your suggestions. We have replaced the word.
Point 9: On line 197, add “after performing Tukey’s HSD test” after “differences”.
Response 9: Thank you for your suggestions. We have added the words.
Point 10: On line 200, add “(P < 0.05)” after “significantly”.
Response 10: Thank you for your suggestions. We have added the words.
Point 11: On line 209, add “(P < 0.05)” after “significantly”.
Response 11: Thank you for your suggestions. We have added the words.
Point 12: On line 213, replace “superscript letters” with “alphabets”.
Response 12: Thank you for your suggestions. We have replaced the word.
Point 13: On line 214, add “after performing Tukey’s HSD test” after “differences”.
Response 13: Thank you for your suggestions. We have added the words.
Point 14: On line 219, add “(P < 0.05)” after “significantly”.
Response 14: Thank you for your suggestions. We have added the words.
Point 15: On line 220, add “(P < 0.05)” after “significantly”.
Response 15: Thank you for your suggestions. We have added the words.
Point 16: On line 229, replace “superscript letters” with “alphabets”.
Response 16: Thank you for your suggestions. We have replaced the word.
Point 17: On line 230, add “after performing Tukey’s HSD test” after “differences”.
Response 17: Thank you for your suggestions. We have added the words.
Point 18: On line 236, add “(P < 0.05)” after “significantly”.
Response 18: Thank you for your suggestions. We have added the words.
Point 19: On line 244, add “significantly (P < 0.05)” before “higher”.
Response 19: Thank you for your suggestions. We have added the words.
Point 20: On line 254, replace “superscript letters” with “alphabets”.
Response 20: Thank you for your suggestions. We have replaced the word.
Point 21: On line 255, add “after performing Tukey’s HSD test” after “differences”.
Response 21: Thank you for your suggestions. We have added the words.
Point 22: On line 259, replace “superscript letters” with “alphabets”, and add “after performing Tukey’s HSD test” after “differences”.
Response 22: Thank you for your suggestions. We have revised the sentence.
Point 23: On line 280, compare the data of qRT-PCR and DEGs statistically following Tukey's HSD and comment on the siginicant difference accordingly.
Response 23: Thank you for your suggestions. We have compared the data of qRT-PCR and DEGs statistically following Tukey's HSD. We have added the comments in lines 293-294 and rebuilt the Figure 10 in the manuscript.
Point 24: On line 292, confirm if P < 0.05.
Response 24: Thank you for your suggestions. We have added the comments in lines 293-294.
Point 25: On line 435, replace “These measures are expressed as mean ± SE” with “The value were expressed as mean ± SE followed by Tukey's HSD test”.
Response 25: Thank you for your suggestions. We have replaced the sentences.
